# Red Meat Intake, Indole-3-Acetate, and *Dorea longicatena* Together Affect Insulin Resistance after Gastric Bypass

**DOI:** 10.3390/nu15051185

**Published:** 2023-02-27

**Authors:** Ana Paula Aguiar Prudêncio, Danielle Cristina Fonseca, Natasha Mendonça Machado, Juliana Tepedino Martins Alves, Priscila Sala, Gabriel R. Fernandes, Raquel Susana Torrinhas, Dan Linetzky Waitzberg

**Affiliations:** 1Laboratory of Nutrition and Metabolic Surgery of the Digestive System, LIM 35, Department of Gastroenterology, Hospital das Clínicas HCFMUSP, Faculdade de Medicina, Universidade de São Paulo, São Paulo 01246-903, SP, Brazil; 2Hospital Sírio Libanês, Department of Medical Clinical Nutrition, Brasilia 70200-730, DF, Brazil; 3Faculty of Nutrition, Universidade São Camilo, São Paulo 04263-200, SP, Brazil; 4Instituto Rene Rachou, Fiocruz Minas, Belo Horizonte 30190-002, MG, Brazil

**Keywords:** Roux-en-Y Gastric bypass, type 2 diabetes, food intake, red meat, tryptophan metabolism, metabolomics, indole-3-acetate, Gut microbiota, insulin resistance

## Abstract

Roux-en-Y Gastric bypass (RYGB) promotes improvement in type 2 diabetes (T2D) shortly after surgery, with metabolic mechanisms yet to be elucidated. This study aimed to investigate the relationship between food intake, tryptophan metabolism, and gut microbiota on the glycemic control of obese T2D women after RYGB surgery. Twenty T2D women who underwent RYGB were evaluated before and three months after surgery. Food intake data were obtained by a seven-day food record and a food frequency questionnaire. Tryptophan metabolites were determined by untargeted metabolomic analysis, and the gut microbiota was determined by 16S rRNA sequencing. The glycemic outcomes were fasting blood glucose, HbA1C, HOMA-IR, and HOMA-beta. Linear regression models were applied to assess the associations between the changes in food intake, tryptophan metabolism, and gut microbiota on glycemic control after RYGB. All variables changed after RYGB (*p* < 0.05), except for tryptophan intake. Jointly, the variation in red meat intake, plasma indole-3-acetate, and *Dorea longicatena* was associated with postoperative HOMA-IR {R^2^ 0.80, R^2^ adj 0.74; *p* < 0.01}. Red meat intake decreased three months after bariatric surgery while indole-3-acetate and *Dorea longicatena* increased in the same period. These combined variables were associated with better insulin resistance in T2D women after RYGB.

## 1. Introduction

Metabolic surgery is a successful treatment for morbid obesity and type 2 diabetes (T2D) [1]. In addition, improvement of T2D after Roux-en-Y Gastric bypass (RYGB) has been noted shortly after surgery, and it is not entirely explained only by weight loss. Many factors have been proposed to elucidate the glycemic improvement of T2D patients following RYGB, including age [2], T2D diagnosis time [2,3], preserved beta pancreatic cell function [4], and preoperative C-peptide levels [5].

Recently, the expansion of metabolomic investigations has raised new evidence linking changes in several metabolites with glycemic enhancement after RYGB, particularly phospholipids, long-chain fatty acids, bile acids, and amino acids [6]. In the field of amino acid research, tryptophan has gained attention, and two studies have demonstrated a relationship between tryptophan metabolites and glucose homeostasis after RYGB [7,8].

Tryptophan, an essential neutral amino acid, is not endogenously synthesized by humans. Dietary intake is necessary, and the main tryptophan food sources include milk and dairy products, eggs, meat, cocoa, and peanuts [9]. In addition to being a serotonin precursor, tryptophan participates in many metabolic pathways and physiological responses. The kynurenine pathway is responsible for approximately 95% of the circulating tryptophan degradation. Metabolites of this pathway are involved in inflammation, and immune response [9], and have been found to be related to diabetes [10]. Kynurenine metabolites can be synthesized by both the host metabolism and gut microbiota. Some bacteria participate in the conversion of tryptophan into kynurenines and derivatives, such as *Clostridium clostrioforme* and the genus Staphylococcus ssp. [11].

Yet another tryptophan metabolic pathway is the indole pathway, triggered in the host, but mainly by gut microbiota bacteria [12] such as *Bifidobacterium longum*, *Bacteroides fragilis*, and *Eubacterium halli* [13]. Metabolites from the indole pathway have a physiological effect by stimulating enteroendocrine cells to secrete glucagon-like secretory peptide-1 (GLP-1) [13]. GLP-1 has a hypoglycemic action due to its insulinotropic properties and the ability to delay apoptosis of pancreatic beta cells [14]. However, to the best of our knowledge, the relationship between indole derivatives and gut microbiota on glycemic improvement after RYGB has not yet been studied in humans.

The intake of fiber and certain food groups seem to have an impact on the circulating metabolites of tryptophan. Qi et al. (2022), when studying 3938 participants from the HCHS/SOL Cohort, observed positive associations between red meat and refined cereal intake with metabolites of the kynurenine pathway, as well as positive associations between the consumption of dietary fiber and indole derivatives [12].

The impact of food intake on tryptophan and glycemic metabolism is already known, as are the changes in gut microbiota after RYGB [15,16,17,18]. Moreover, tryptophan metabolites produced both by host and gut bacteria appear to be related to the physiopathology of obesity and T2D as well as to the T2D improvement after RYGB [7,8]. However, the effect of changes in food intake, gut microbiota, and tryptophan metabolites on glycemic homeostasis is not yet thoroughly understood. Thus, we aimed to investigate the relationship between food intake, tryptophan metabolism, and gut microbiota on glycemic control in obese T2D women after RYGB surgery. The impact of our findings includes new insight into the knowledge of T2D relief after RYGB and raises possible future therapeutic targets.

## 2. Materials and Methods

### 2.1. Study Design and Subjects

This is a single-institution study, approved by the local ethics committee (CaPPesq 4.019.801), which was a part of the SURmetaGIT trial [19] registered with www.ClinicalTrials.gov (NCT01251016; 8 December 2015). Written informed consent was acquired from all participants before the beginning of the study. All protocol interventions were performed following the Declaration of Helsinki guidelines.

Women with obesity-related T2D and candidates for RYGB were recruited from the Surgical Gastroenterology Department of the Hospital das Clínicas of the University of São Paulo, School of Medicine. Data collection was performed between February 2011 and December 2014. Inclusion criteria were as follows: women (18–60 years) with body mass index (BMI) ≥ 35 kg/m^2^, associated with T2D diagnosis (fasting blood glucose [FBG] ≥ 126 mg/dL and glycated hemoglobin [HbA1C] ≥ 6.5%) and/or use of oral hypoglycemic agent [20]. Patients with recent participation in other interventional study protocols, or with *Helicobacter pylori* infection, thyroid, or hepatic diseases, under insulin therapy or antibiotic, probiotic, and prebiotic use in the month preceding fecal sample collection were excluded. The RYGB procedure was previously described in the SURMETAGIT protocol [19]. Briefly, open RYGB without silicon rings with a standardized length of biliary-pancreatic limb (50–60 cm) and alimentary limb (100–120 cm) were performed. Food intake surveys, plasma, and fecal samples were collected before and three months after RYGB. Plasma samples were obtained by centrifugation (2800 rpm at 4 °C for 10 min) of blood samples collected after a 12-h fast in ethylenediaminetetraacetic acid (EDTA)-containing tubes (Complete™ mini, EDTA free, Lifescience, Roche Diagnostics Corporation, Indianapolis, IN, USA). These samples were maintained at −80 °C until biochemical and metabolomic analysis. The fecal samples were self-collected by the patients at home, by using a specific specimen collection system (Commode Specimen; Fisher Scientific, Ottawa, ON, Canada). After collection, fecal samples were immediately frozen at −20 °C and transported under controlled temperature to our laboratory, where they were immediately aliquoted (100 mg) into cryogenic vials (without thawing) and stored at −80 °C until gut microbiota evaluations.

### 2.2. Food Intake

Food intake data were obtained by a seven-day food record (7dFR) and a food frequency questionnaire (FFQ), applied one week before both stool sample collections, as previously described by our group [19]. Briefly, food reported in 7dFR was registered in cooking units (such as tablespoons), guided by illustrations from a manual offered to all patients [21]. The research team converted these units to grams or milliliters after standardization [22]. Energy intake, macronutrients, and total fiber were determined by Virtual Nutri Plus^®^ software, which includes the Brazilian Table of Food Composition (TACO) [23] and the Table of Food Composition: Support to Nutritional Decision [24]. In the present study, we estimated the tryptophan, usual energy, and nutrient intake, as well as the food groups of interest. Tryptophan intake was determined from food sources available in the Brazilian Food Composition Table [23] and the Food Composition Table of the United States Department of Agriculture (USDA) [25]. The Multiple Source Method (MSM) was applied to estimate the usual energy and nutrient intakes through the online platform [26]. All nutrients were adjusted for total energy intake by the residue method [27]. From the FFQ, we determined the daily intake of two food groups: red meat (beef and viscera, i.e., liver, heart, and kidney) and refined cereals (rice, pasta, bread, cakes, salty crackers, and cookies).

### 2.3. Tryptophan Metabolites

Tryptophan Metabolites were identified in plasma samples by untargeted metabolomic analysis, performed previously by our group at the NIH West Coast Metabolomics Center (WCMC), located at the Genome Center at the University of California, Davis (United States of America) [28]. Seven tryptophan metabolites were captured by mass spectrophotometry using a multiplatform approach combining three analytical platforms: 6530 Accurate-Mass Q-TOF LC/MS e Agilent 1290 Infinity II LC System (Agilent Technologies Ò), a high-performance liquid chromatography (HPLC)—TOF tandem mass spectrometer (MS/MS) method with hydrophilic interaction column (HILIC)—for polar compounds—and the charged hybrid surface column (CSH) for non-polar compounds; and TOF-coupled gas chromatography on the Agilent 6890 GC Pegasus III TOF MS instrument.

The analytical variation and the reproducibility of the profiles were verified during the analyses with standard technic to guarantee the consistency of the results and ensure the instrument itself did not cause large random or systematic deviations from the data obtained during sample acquisition. This was accomplished using a mixture of reference molecules that covered all chemical classes of the metabolites identified in typical analyses (quality control samples).

The raw data obtained were converted using the Analysis Base File Converter software (Reifycs Inc., Tokyo, Japan). Data from metabolites identified by LC-MS were processed by the free MS-DIAL software developed at WCMC (http://prime.psc.riken.jp/Metabolomics_Software/; 13 November 2017). Primary metabolites analyzed by GC-MS spectra were processed based on BinBase data. The results were filtered based on multiple parameters to exclude inconsistent peaks. All BinBase entries were compared to mass spectra from the FiehnLib library with 1200 authentic spectra using retention index information and mass spectra or from library 11 from the National Institute of Standards Technology (NIST).

Data obtained from the plasma samples were reported as the height of quantitative peaks, normalized by the sum of the intensity of all identified metabolites (mTIC), and used for further statistical analyses.

The fold change (FC) was applied to determine the relative changes and to describe its effect size and the direction of metabolite changes. The calculation consisted of the ratio between the postoperative/preoperative mean, and values < 1 were converted and added with a negative sign (−).

### 2.4. Gut Microbiota (GM)

The GM evaluations were previously developed at MetaGenoPoliS at Jouy-en-Josas, France (http://www.mgps.eu; 5 October 2017) by obtaining fecal DNA and amplifying the V4 region of the 16S rRNA gene, as detailed in the International Human Microbiome Standards (IHMS) SOP06 (http://www.microbiome-standards.org; 11 October 2017) and documented by our group [29]. In the present study, the bioinformatic analysis of the 16S rRNA data was conducted by amplicon sequence variants (ASV) analysis to achieve better resolution for bacteria identification. ASV analysis was carried out at the Bioinformatics Platform in Rene Rachou Institute, Fiocruz Minas (Belo Horizonte, MG, Brazil. Briefly, raw sequence reads of the 16S rRNA gene analysis and ASV calling were performed using the DADA2 [30]. The primers used in the amplification were removed, and sequences with more than two expected errors were discarded. The remaining sequences were used to train an error identification and correction model. The forward and reverse readings, already corrected, were concatenated to form ASVs, remove chimeric sequences, and quantify ASVs. Each ASV had its taxonomic classification assigned by the TAG.ME package [31], using the specific model for the amplicon that corresponds to the V4 region, according to the Silva database [32]. The alpha diversity indexes of the microbial communities (Simpson, Shannon, observed species, Fisher, Ace, and Chao1) were calculated using the Phyloseq package (1.40.0) [33].

### 2.5. Outcomes

Biomarkers of glycemic control were used as outcomes in statistical regression models. Systemic concentrations of fasting blood glucose (FBG), glycated hemoglobin (HbA1c), and insulin were measured by an enzymatic method (glucose), liquid chromatography (HbA1c), and electrochemiluminescence (insulin), at the Central Laboratory Division of HC-FMUSP, as previously described by our group [19]. Additionally, the Homeostasis Assessment Model (HOMA) was applied to determine the degree of insulin resistance (IR) and the functional capacity of the pancreatic beta cells (Beta) [34].

### 2.6. Statistical Analysis

Continuous variables are presented as the mean and standard deviation or median and interquartile range, while categorical variables are presented as absolute and relative frequencies. The normality of continuous variables was assessed using the Shapiro–Wilk test.

Differences in the relative abundance of gut microbiota bacteria between the periods studied (preoperative and three months postoperatively) were determined using the Phyloseq package (1.40.0) [33]. Comparisons between each period of the variables of food consumption and metabolites were performed by paired t-test or Wilcoxon test. A significance level of 5% (*p* < 0.05) was adopted for these analyses.

Associations between the independent variables—food consumption, metabolites, and intestinal microbiota—individually or in groups—and the variables related to glycemic control (outcomes) were assessed by simple and multiple linear regression, respectively. For GI variables, we included only bacteria that vary between periods (*p* < 0.05) in the regression models. We adopted a significance level of 5% (*p* < 0.05) to evaluate the glycemic outcomes affected by the independent variables—alone or within its groups (food consumption, metabolites, and intestinal microbiota).

To investigate the effect of combined variables from the different groups (food consumption, metabolites, and intestinal microbiota) on glycemic outcomes we performed multiple regression models. Initially, we performed linear regression models to pre-select independent variables that were associated with glycemic outcomes individually or in groups (*p* ≤ 0.1) [35]. After this previous selection of independent variables, the olsrr package [36] was used to find the best combination of two or more variables from different groups that could explain the dependent variable in question (FBG, HbA1c, HOMA-IR, and HOMA-Beta). The best subset of predictors was estimated for each dependent variable. The selection of these subsets was based on the F statistic values, the significance of the estimates, adjusted R^2^, mean square error, Masllow’s Cp, and Akaike’s information criterion. After finding the best models (*p* < 0.05), they were tested for normality, heteroscedasticity, multicollinearity, and autocorrelation. Finally, based on the best model found to explain the dependent variables in question (FBG, HbA1c, HOMA-IR, and HOMA-Beta), we assumed a significance of *p* < 0.05 to determine significant effects with the intervention.

The comparison tests were performed with the Statistical Package for Social Science (SPSS) program, version 12.0. For the regression models, specific packages of the R software (version 4.2) were used.

## 3. Results

### 3.1. Patient’s Descriptive Data

Twenty women were included in the study. At baseline, participants were 47 ± 6.5 years old, with a BMI of 46.5 ± 5.9 kg/m^2^, had all glycemic biomarkers compatible with T2D, and used at least one oral hypoglycemic agent. As shown in Table 1, all anthropometric and biochemical data changed after RYGB, except for High-Density Lipoprotein Cholesterol (HDL-c) levels. Biomarkers of glycemic control indicate that RYGB promoted an improvement of T2D. Only two participants maintained the use of oral hypoglycemic agents at three months after surgery.

### 3.2. Food Intake

Three months after RYGB, all participants presented a reduction in their intake of energy, macronutrients, red meat, and refined cereals. However, probably due to the changes in food choices after surgery when protein food groups such as milk and eggs were preferred over meat, tryptophan intake did not differ between the two recorded periods (Table 2). In addition, 35% of patients reported albumin supplement intake. These food groups and albumin supplements are sources of tryptophan.

### 3.3. Tryptophan Metabolites

As shown in Table 3, RYGB promoted changes in plasma tryptophan metabolites; N-acetyl-serotonin and indole-3-acetate increased after surgery. Conversely, only anthranilic acid decreased in the same period. These changes indicate a metabolic effect of RYGB on the three major tryptophan pathways (Figure 1).

### 3.4. Gut Microbiota (GM)

RYGB did not change the GM alpha diversity (Appendix A) but affected the GM composition. We observed changes in 27 ASV bacteria taxa between pre- and postoperative time points, in which 3 were reduced and 24 increased. These represent 5 differences in bacteria phyla, and 22 differences in species, as described in Table 4. As shown in Figure 2, among the most prevalent gut bacteria phyla, only Verrucomicrobia and Fusobacteria abundance increased three months after RYGB (vs. preoperative).

### 3.5. Regression Models

Univariate and multivariate regression models showed associations between the variation (∆ postoperative—preoperative) in food intake, tryptophan metabolites, and gut microbiota with surrogate markers of glycemic control after bariatric surgery (Appendix A). The variation in tryptophan metabolites, individually or together, did not affect any postoperative glycemic biomarkers (*p* > 0.05). The variation in Fusobacterium nucleatum (sq381) was directly associated with postoperative glycemia {0.09 (0.02, 0.16); *p* = 0.05}.

For food intake, a variation in red meat intake was positively associated with postoperative glycemia {0.10 (0.03,0.17); *p* = 0.03} and HOMA-IR {0.01 (0.005, 0.01); *p* = 0.002}. Therefore, the greater reduction in red meat intake after RYGB was directly associated with postoperative glycemia and HOMA-IR. In addition, multiple linear regression revealed that the variation in protein {−0.05 (−0.09, −0.02); *p* = 0.03} and red meat {0.01 (0.01, 0.01); *p* = 0.0001} intake, individually and together, affected HOMA-IR (R^2^ 0.68, adjusted R^2^ 0.63; *p* < 0.01). Individually, a variation in protein intake was inversely associated with HOMA-IR, thus, the smaller reduction in protein intake after RYGB, the better improvement of postoperative insulin resistance.

The only significant model that assembled variables from each group (food intake, metabolites, and gut microbiota) on dependent glycemic variables was the HOMA-IR model (Table 5). Together, the reduction in red meat intake {0.01 (0.005, 0.01); *p* = 0.0003}, an increase in plasma indole-3-acetate {−0.001 (−0.001, −0.0001); *p* = 0.06}, and Dorea longicatena (sq1408) {0.03 (0.01, 0.06); *p* = 0.06} were able to explain the improvement of postoperative insulin resistance (HOMA-IR) {R^2^ 0.80, R^2^ adj 0.74; *p* < 0.01}.

## 4. Discussion

Our study showed that RYGB promoted the glycemic improvement of all biomarkers evaluated. At three post-operative months, all patients achieved targets for fasting blood glucose, %HbA1c, and HOMA-IR. We also demonstrated that food intake, gut microbiota, and tryptophan metabolite changes affected glucose homeostasis after RYGB.

Regarding food intake, we observed changes in protein source choices during the postoperative period, marked by a decrease in red meat intake. Reduction in red meat consumption is very common after RYGB as it is reported to be a less tolerated food after the anatomical changes induced by the surgery [37,38,39]. This intolerance may be a consequence of changes in protein digestion caused by the reduction in pepsin synthesis in the gastric pouch, as well as inadequate chewing and increased satiety due to changes in the gut hormones involved in gastric motility and gastric acid secretion [37,38].

Variation in red meat intake was positively associated with postoperative glycemia and HOMA-IR, while protein intake variation was inversely related to postoperative HOMA-IR. Since red meat intake was reduced after RYGB, postoperative glycemia and insulin resistance could be partially explained by a significant decrease in red meat intake. Literature data regarding the association between red meat intake and T2D risk are conflicting. While observational studies have suggested that a higher red meat intake increases the risk of T2D incidence [40,41,42], randomized controlled trials do not confirm associations of red meat intake patterns with the glycemic biomarkers of T2D patients [43]. These divergent results could be attributed to confounding factors that affect glycemic homeostasis and are usually associated with red meat intake, such as alcohol consumption, sedentary lifestyle, and low fiber intake [43]. Nevertheless, red meat compounds seem to affect both beta pancreatic cell function and hepatic insulin extraction by increasing reactive oxygen species and hepatic glucose synthesis, respectively [44].

Decreased protein intake after RYGB was associated with improved postoperative insulin resistance. The anatomic gastrointestinal changes promoted by RYGB increase the availability of partially-digested nutrients to intestinal microbiota, including proteins [45]. When they reach the large intestine, partially digested proteins induce proteolytic bacteria growth, increasing the potential to synthesize pro-inflammatory metabolites, such as hydrogen sulfide (H2S) and trimethylamine-N-oxide (TMAO) [46]. However, a dietary source of protein may influence the intestinal microbiota composition [47]. In this context, reduced red meat intake and a preference for milk, eggs, and albumin supplements seem to decrease some fecal pro-inflammatory bacterial species [48], shifting the gut microbiota composition to more beneficial bacteria [49], which may interfere with glucose homeostasis [50].

*Fusobacterium (F.) nucleatum* was increased after surgery and its changes affected postoperative glycemia improvement. *F. nucleatum* is an anaerobic bacterium that engages in diverse interactions with other microorganisms and humans and can be both beneficial and detrimental in nature [51]. To the best of our knowledge, this is the first study to report the association between *F. nucleatum* and glycemia levels after RYGB. Thus, the mechanisms involved in this finding remain unclear. In our study, increased *Akkermansia Muciniphila* and reduced *Faecalibacterium prausnitzii* did not affect T2D improvement after RYGB. This is notable as both bacteria have been associated with better metabolic biomarkers in healthy and T2D individuals [52,53,54,55].

Changes in tryptophan metabolites induced by RYGB included decreased anthranilic acid and an increase in both N-acetyl-serotonin and indole-3-acetate; these alterations might indicate a downstream change in the kynurenine pathway, with a shift towards the serotonin and indole pathways, respectively. This redirection of tryptophan pathways may be due to a low-grade inflammation reduction [56] and gut microbiota changes after RYGB [57]. Tryptophan is converted to kynurenine by pro-inflammatory and stress hormones and activation of the IDO and TDO enzymes, respectively [58]. The new scenario with fewer pro-inflammatory signals may reduce the conversion of tryptophan to kynurenine so more tryptophan is available for the indole and serotonin pathways [59]. Furthermore, indole producer bacteria have been shown to increase after surgery, such as *Dorea longicatena*, and *Akkermansia muciniphila* [11].

Moreover, changes in tryptophan metabolites after bariatric surgery have been described. Christensen et al. 2018 [60] showed a reduction in plasma tryptophan, kynurenine, and all kynurenine metabolites, except anthranilic acid, three months after bariatric surgery (sleeve gastrectomy and biliopancreatic diversion with duodenal switch). Favennec et al. 2016 [7] also observed a reduction in plasma tryptophan, kynurenine, and all kynurenine metabolites, and an increase in serotonin one year after sleeve gastrectomy and RYGB. Kwon et al. 2021 [61] reported an increase in indoxyl sulfate but no changes in indole-3-acetate and indole-3-pyruvate three months after sleeve gastrectomy. Yeung et al. 2022 found reduced levels of tryptophan, kynurenic acid, and xanthurenic acid three months after RYGB [8].

In disagreement with these four studies, we did not find an association in the variation of tryptophan metabolites individually on any glycemic biomarkers after bariatric surgery. However, the variation of indole-3-acetate together with red meat intake and *Dorea longicatena* was able to potentially explain the improvement of postoperative insulin resistance. *Dorea longicatena* is a producer of indole-3-acetate [11], and both were increased after RYGB. Indole-3-acetate activates aryl hydrocarbon receptors, reducing inflammation and insulin resistance [62]. In addition, indole derivates stimulate insulin secretion through the GLP-1 release [58]. Furthermore, higher plasmatic levels of indole-3-acetic acid have been associated with lower insulin resistance after sleeve gastrectomy [61]. Despite that, to the best of our knowledge, this is the first study to investigate the interaction of food intake, tryptophan metabolism, and gut microbiota variables on glycemic homeostasis, in addition to reporting that alterations of red meat intake, indole-3-acetate, and *Dorea longicatena* together affects insulin resistance after RYGB.

As a limitation of this present study, the findings do not eliminate the potentiality of other tryptophan metabolites to also affect insulin resistance and glycemic biomarkers after RYGB. In addition, we could only include a small number of participants. Furthermore, some researchers have shown that the experimental absence of gut microbiota could change the concentration of tryptophan in plasma, leading to a reduction in the kynurenine-to-tryptophan ratio [63]. Even if tryptophan metabolism by gut microbiota seems relatively simple at the molecular level, and its metabolic transformation of tryptophan into other metabolites seems undeniable, it is challenging to determine which metabolite each bacterium can produce due to the high diversity and complexity of the microbiome [64]. Thus, we encourage further investigations with a greater panel of indole and kynurenine derivates and a larger number of patients to validate our findings.

## 5. Conclusions

Early after RYGB, in obese T2D women, there are changes in tryptophan metabolism, food intake, and gut microbiota. Some of these changes can be related to glycemic homeostasis. Together, alterations of red meat intake, indole-3-acetate, and *Dorea longicatena* seem to improve the surrogate markers associated with insulin resistance at the three-month post-operatory period.

## Figures and Tables

**Figure 1 nutrients-15-01185-f001:**
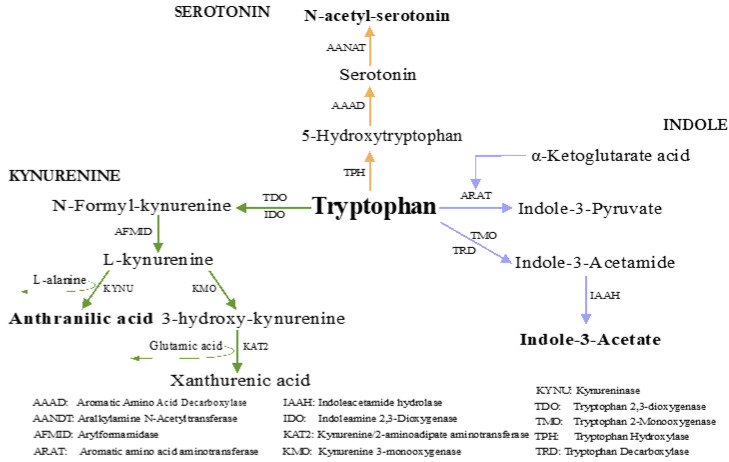
Schematic representation of the identified metabolites separated by the three major tryptophan pathways. The metabolites changed three months after RYGB are highlighted in bold. Statistical significance *p* < 0.05.

**Figure 2 nutrients-15-01185-f002:**
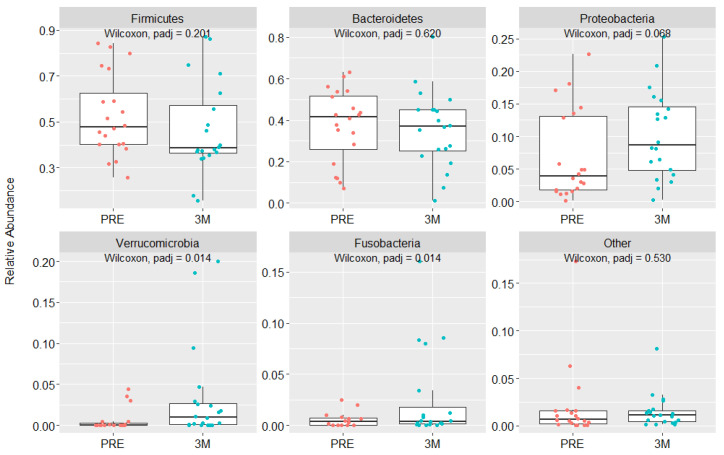
Relative abundance (%) of changes in gut bacteria phyla before and three months after RYGB. Statistical significance *p* < 0.05.

**Table 1 nutrients-15-01185-t001:** Characterization of the participants according to clinical and biochemical variables before and three months after RYGB.

Variables	Preoperative	Postoperative (3 Months)	*p*-Value ^1^
Age, years	46.6 ± 6.5	NA ^2^	
Albumin supplement, *n* (%)	0 (0)	7 (35)	
Oral hypoglycemic			
Metformin, *n* (%)	5 (25)	1 (5)	
Glibenclamide, *n* (%)	1 (5)	0 (0)	
Acarbose, *n*, (%)	0 (0)	0 (0)	
Glicazide, *n* (%)	0 (0)	0 (0)	
Multimedication, *n* (%)	14 (70)	1 (5)	
Antihyperlipidemic agents, *n* (%) ^3^	9 (45)	2 (10)	
Anthropometric data			
Weight, kg	114.4 ± 16.4	93.6 ± 12.9	0.000
BMI ^4^, kg/m^2^	46.5 ± 5.9	38.0 ± 4.6	0.000
EWL ^5^, %	NA^2^	33.7 ± 5.5	
Biochemical data			
FBG ^6^, mg/dL	219.6 ± 77.2	100.6 ± 19.9	0.000
HbA1c, %	9.3 ± 1.7	6.1 ± 0.4	0.000
Insulin, µU/mL	15.7 (6.8)	7.8 (4.9)	0.001
HOMA ^7^-IR	7.9 (5.7)	1.9 (1.2)	0.000
HOMA ^7^ Beta	47.1 (53.3)	90.8 (72.2)	0.019
Triglycerides, mg/dL	136 (64)	103 (25)	0.012
HDL-c ^8^, mg/dL	45.3 ± 10.4	42.9 ± 10.2	0.254

^1^ paired *t*-test or Wilcoxon test, statistical significance *p* < 0.05. ^2^ NA = not applicable. ^3^ Statins and fibrates. ^4^ BMI = Body Mass Index. ^5^ EWL = excess weight lost. ^6^ FBG = fasting blood glucose. ^7^ HOMA = Homeostatic model assessment. ^8^ HDL-c = High-Density Lipoprotein Cholesterol. Data are shown as the mean ± standard deviation or median (interquartile range).

**Table 2 nutrients-15-01185-t002:** Food intake variables before and three months after RYGB.

Variables	Preoperative	Postoperative (3 Months)	*p*-Value ^1^
Energy, kcal	1.696.9 ± 375.4	973.2 ± 211.6	0.000
Protein, g	68.7 ± 11.9	47.9 ± 13.9	0.006
Protein, %	16.7 ± 2.3	22.2 ± 5.6	0.000
Carbohydrates, g	209.8 ± 42.4	107.7 ± 25.5	0.000
Carbohydrates, %	50.9 ± 3.7	44.8 ± 5.5	0.000
Lipids, g	62.1 ± 15.5	37.7 ± 7.7	0.000
Lipids, %	33.6 ± 3.5	35.4 ± 3.5	0.083
Fiber, g	14.8 ± 5.3	9.1 ± 3.4	0.000
Tryptophan, mg	232.1 (66.1)	230.5 (170.0)	0.299
Tryptophan, mg/kg	2.0 (1.0)	2.4 (2.3)	0.027
Red meat, g	51.4 (103.0)	28.9 (26.8)	0.015
Refined cereals, g	361.8 (341.2)	59.2 (60.4)	0.000

^1^ paired t-test or Wilcoxon test, statistical significance *p* < 0.05. Data are shown as the mean ± standard deviation or median (interquartile range).

**Table 3 nutrients-15-01185-t003:** Changes in tryptophan metabolites three months after RYGB.

Plasma Metabolites	Fold Change	*p*-Value ^1^
Tryptophan	−1.62	0.112
N-acetyl-serotonin	1.32	0.008
Indole-3-acetate	1.77	0.016
Glutamic acid	−1.07	0.372
L-alanine	1.06	0.528
α-ketoglutarate acid	1.14	0.157
Anthranilic acid	−1.87	0.004

^1^ Wilcoxon test, statistical significance *p* < 0.05.

**Table 4 nutrients-15-01185-t004:** Relative abundance (%) of gut bacteria species before and three months after RYGB.

Bacterial Taxonomy							Preoperative	Postoperative	*p*-adj ^1^
sq	Phylum	Class	Order	Family	Genus	Species		(3 Months)	
sq18	Firmicutes	Clostridia	Clostridiales	Veillonellaceae	Veillonella	Veillonella parvula	0.103 ± 0.137	1.827 ± 4.187	0.0002
sq28	Firmicutes	Bacilli	Lactobacillales	Streptococcaceae	Streptococcus	Streptococcus salivarius	0.240 ± 0.562	1.385 ± 2.216	0.0037
sq58	Firmicutes	Clostridia	Clostridiales	Lachnospiraceae	Lachnoclostridium	Clostridium clostridioforme	0.115 ± 0.242	0.940 ± 3.278	0.0421
sq96	Firmicutes	Clostridia	Clostridiales	Lachnospiraceae	Blautia	Blautia luti	0.538 ± 0.669	0.141 ± 0.222	0.0020
sq80	Firmicutes	Negativicutes	Selenomonadales	Veillonellaceae	Veillonella	Veillonella parvula	0.143 ± 0.568	0.482 ± 0.887	0.0069
sq93	Firmicutes	Bacilli	Lactobacillales	Streptococcaceae	Streptococcus	Streptococcus parasanguinis	0.062 ± 0.185	0.456 ± 0.606	0.0045
sq315	Firmicutes	Negativicutes	Selenomonadales	Veillonellaceae	Veillonella	Veillonella parvula	0.000 ± 0.000	0.079 ± 0.180	0.0225
sq311	Firmicutes	Bacilli	Bacillales	Family XI	Gemella	Gemella haemolysans	0.006 ± 0.010	0.069 ± 0.097	0.0064
sq338	Firmicutes	Clostridia	Clostridiales	Clostridiaceae 1	Sarcina	Clostridium tarantellae	0.005 ± 0.020	0.064 ± 0.099	0.0167
sq380	Firmicutes	Clostridia	Clostridiales	Lachnospiraceae	Lachnoclostridium	Clostridium lavalense	0.011 ± 0.019	0.047 ± 0.092	0.0033
sq421	Firmicutes	Bacilli	Lactobacillales	Carnobacteriaceae	Granulicatella	Granulicatella adiacens	0.003 ± 0.009	0.039 ± 0.056	0.0253
sq768	Firmicutes	Clostridia	Clostridiales	Lachnospiraceae	Eisenbergiella	Eisenbergiella tayi	0.002 ± 0.007	0.018 ± 0.038	0.0209
sq649	Firmicutes	Clostridia	Clostridiales	Ruminococcaceae	Faecalibacterium	Faecalibacterium prausnitzii	0.013 ± 0.021	0.003 ± 0.006	0.0440
sq691	Firmicutes	Clostridia	Clostridiales	Lachnospiraceae	Lachnoclostridium	Clostridium clostridioforme	0.003 ± 0.008	0.011 ± 0.019	0.0445
sq1127	Firmicutes	Bacilli	Lactobacillales	Enterococcaceae	Enterococcus	Enterococcus faecalis	0.001 ± 0.004	0.004 ± 0.008	0.0120
sq1542	Firmicutes	Clostridia	Clostridiales	Lachnospiraceae	Oribacterium	Oribacterium parvum	0.000 ± 0.000	0.004 ± 0.006	0.0223
sq1408	Firmicutes	Clostridia	Clostridiales	Lachnospiraceae	Ruminococcus torques group	Dorea longicatena	0.000 ± 0.000	0.003 ± 0.005	0.0360
sq1316	Firmicutes	Negativicutes	Selenomonadales	Veillonellaceae	Dialister	Dialister invisus	0.000 ± 0.001	0.002 ± 0.003	0.0440
sq1698	Firmicutes	Clostridia	Clostridiales	Family XIII	Mogibacterium	Mogibacterium vescum	0.000 ± 0.000	0.002 ± 0.003	0.0225
sq2371	Firmicutes	Clostridia	Clostridiales	Peptostreptococcaceae	Clostridioides	Clostridioides difficile	0.001 ± 0.002	0.000 ± 0.000	0.0213
sq368	Fusobacteria	Fusobacteriia	Fusobacteriales	Fusobacteriaceae	Fusobacterium	Fusobacterium periodonticum	0.002 ± 0.007	0.055 ± 0.099	0.0016
sq381	Fusobacteria	Fusobacteriia	Fusobacteriales	Fusobacteriaceae	Fusobacterium	Fusobacterium nucleatum	0.000 ± 0.000	0.039 ± 0.079	0.0025
sq612	Fusobacteria	Fusobacteriia	Fusobacteriales	Fusobacteriaceae	Fusobacterium	Fusobacterium nucleatum	0.000 ± 0.000	0.019 ± 0.032	0.0092
sq1129	Fusobacteria	Fusobacteriia	Fusobacteriales	Fusobacteriaceae	Fusobacterium	Fusobacterium nucleatum	0.000 ± 0.000	0.004 ± 0.009	0.0143
sq3	Proteobacteria	Gammaproteobacteria	Enterobacteriales	Enterobacteriaceae	Escherichia-Shigella	Escherichia coli	1.255 ± 2.522	6.397 ± 6.410	0.0003
sq647	Proteobacteria	Gammaproteobacteria	Betaproteobacteriales	Neisseriaceae	Neisseria	Neisseria flavescens	0.000 ± 0.000	0.008 ± 0.014	0.0141
sq15	Verrucomicrobia	Verrucomicrobiae	Verrucomicrobiales	Akkermansiaceae	Akkermansia	Akkermansia muciniphila	0.010 ± 0.018	2.116 ± 4.761	0.0280

^1^ *p* adjusted: Wilcoxon test, statistical significance *p* < 0.05. Data are shown as the mean ± standard deviation.

**Table 5 nutrients-15-01185-t005:** Multiple regression model with the effect of variation (∆ postoperative—preoperative) of food intake, tryptophan metabolites, and gut microbiota (independent variables) on postoperative HOMA-IR (dependent variable).

Variables	HOMA-IR ^1^
Food intake	
Red meat (g)	0.01 (0.005, 0.01) *p* = 0.0003
Metabolites	
Indole-3-acetate	−0.001 (−0.001, −0.0001) *p* = 0.06
Gut Microbiota	
*Dorea longicatena* (sq1408)	0.03 (0.01, 0.06) *p* = 0.06

^1^ Multivariate regression model: HOMA-IR R^2^ 0.80; R^2^ adj 0.74 (*p* < 0.01). Statistical significance *p* < 0.05.

## Data Availability

Nucleotide sequence data used for this study are deposited in the Sequence Read Archive (SRA) of the NCBI under the accession number SRP113514.

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
