# Peer review of "Red Meat Intake, Indole-3-Acetate, and *Dorea longicatena* Together Affect Insulin Resistance after Gastric Bypass"

_nutrients, 2023, doi:10.3390/nu15051185_

Round 1
Reviewer 1 Report
Dear authors,
I have studied with great interest the manuscript “Red meat intake, indole-3-acetate, and Dorea longicatena together affect insulin resistance after gastric bypass”. The work presented is original and may have potential relevance in the future.
1. In the introduction section the impact of the present work could be more
elaborated.
2. Provide the statistical significance for all figures and Tables.
3. Spell out the terms when they are first time in the manuscript like HbA1C, HOMA-IR
4. A few places where references for important points that substantiate the logic or discussion of results are lacking.
5. Include citations related to the previous study and methods followed in the
manuscript.
Reviewer 2 Report
The manuscript titled "Red meat intake, indole-3-acetate, and Dorea longicatena together affect insulin resistance after gastric bypass" demonstrated that after RYGB surgery, the reduction of red meat intake, Indole-3-acetate and Dorea longicatena increased are related to better insulin resistance in women with type 2 diabetes. The information in this article may be helpful to study the mechanism of type 2 diabetes relief after RYGB surgery. I have a few comments that may help the authors improve the manuscript.
1. Figure 2 legend font is not uniform and needs to be modified.
2. What does T2DM mean in the manuscript, please correct it.
3. Line 208, the significance level of the manuscript is 5% (p < 0.05), is this sentence wrong?
4. Table 4, some formats in the table need to be adjusted. There are some other formats in the manuscript that need to be modified, such as line 288, 301.
5. The manuscript shows the data statistics of patients 3 months after surgery, Are there any longer data statistics?
Round 2
Reviewer 2 Report
Accept this revised version
